# Increase in *TPSB2* and *TPSD1* Expression in Synovium of Hip Osteoarthritis Patients Who Are Overweight

**DOI:** 10.3390/ijms241411532

**Published:** 2023-07-16

**Authors:** Maho Tsuchiya, Kensuke Fukushima, Ken Takata, Yoshihisa Ohashi, Katsufumi Uchiyama, Naonobu Takahira, Hiroki Saito, Ayumi Tsukada, Gen Inoue, Masashi Takaso, Kentaro Uchida

**Affiliations:** 1Department of Orthopedic Surgery, Kitasato University School of Medicine, 1-15-1 Kitasato, Minami-ku, Sagamihara City 252-0374, Japan; 09.ma.10.ho@gmail.com (M.T.); kenfu@r4.dion.ne.jp (K.F.); kentakata41@yahoo.co.jp (K.T.); 44134413oo@gmail.com (Y.O.); kgka.condition-green@xd6.so-net.ne.jp (H.S.); amidesutarere9010@yahoo.co.jp (A.T.); ginoue@kitasato-u.ac.jp (G.I.); mtakaso@kitasato-u.ac.jp (M.T.); 2Department of Patient Safety and Healthcare Administration, Kitasato University School of Medicine, 1-15-1 Kitasato, Minami-ku, Sagamihara City 252-0374, Japan; katsufu@cf6.so-net.ne.jp; 3Department of Rehabilitation, Kitasato University School of Allied Health Sciences, 1-15-1 Kitasato, Minami-ku, Sagamihara City 252-0373, Japan; takahira@med.kitasato-u.ac.jp

**Keywords:** hip osteoarthritis, mast cell, obesity, tryptase

## Abstract

While research suggests that increasing body mass index (BMI) is a risk factor for hip osteoarthritis (HOA), the mechanisms of this effect are not fully understood. Tryptases are among the main proteases found in mast cells (MCs) and contribute to OA pathology. *TPSB2*, which encodes β-tryptase, is increased in the synovium of overweight and obese knee OA patients. However, it remains unclear whether tryptase in the synovium of HOA is increased with increasing BMI. Here, we investigated tryptase genes (*TPSB2* and *TPSD1*) in the synovium of overweight HOA patients. Forty-six patients radiographically diagnosed with HOA were allocated to two groups based on BMI, namely normal (<25 kg/m^2^) and overweight (25–29.99 kg/m^2^). *TPSB2* and *TPSD1* expression in the synovium of the two groups was compared using real-time polymerase chain reaction. To compare *TPSB2* and *TPSD1* expression in MCs between the groups, we isolated the MC-rich fraction (MC-RF) and MC-poor fraction (MC-PF), extracted using magnetic isolation. *TPSB2* and *TPSD1* expression was increased in the overweight group compared with the normal group. Expression of both genes in the MC-RF was significantly higher than that in MC-PF in both groups. However, *TPSB2* and *TPSD1* expression levels in the MC-RF did not differ between the groups. Tryptase genes were highly expressed in the synovium of overweight HOA patients. Further investigation to reveal the role of tryptase in the relationship between increasing BMI and HOA pathology is required.

## 1. Introduction

Osteoarthritis is a degenerative and whole-joint disease that causes pain, stiffness, and reduced mobility. Epidemiological studies suggest that being overweight or obese is a significant risk factor for developing osteoarthritis (OA), particularly in the knee and hip [1,2,3,4,5]. Many studies have investigated the mechanism of knee OA (KOA). Synovial tissue (ST) plays a pivotal role in the inflammatory processes and is involved in OA onset, progression, and pain [6,7]. In contrast, the mechanism of the effect of obesity on ST and the exacerbation of hip osteoarthritis is not completely understood.

Mast cells (MCs) are responsible for the production of inflammatory cytokines and their release in conditions such as asthma and allergies [8,9]. Accumulated evidence suggests that MCs contribute to synovial inflammation and OA progression [10,11,12]. We previously reported increased expression of MC markers in the ST of overweight and obese patients with KOA [13,14,15]. Nevertheless, it remains unclear whether MC are increased in the ST of hip osteoarthritis (HOA) with increased body mass index (BMI).

It is well known that tryptases are among the most abundant proteases found in MC [9,16,17]. An increase in tryptase enzyme activity in the synovial extract of KOA patients compared with healthy subjects has been documented [18]. In human MCs, at least four tryptase genes are expressed, *TPSAB1*, *TPSB2*, *TPSD1*, and *TPSG1*, which encode soluble (*TPSAB1*, *TPSB2*) and membrane-anchored (*TPSD1* and *TPSG1*) forms of tryptase [19]. *TPSB2* encodes β-tryptase, the main type released during degranulation in MCs. β-Tryptase is also involved in the production of inflammatory cytokines in inflammatory conditions and inflammatory diseases, as well as being the most abundant protein in human MCs [16,17]. In contrast to *TPSB2*, *TPSD1* encodes δ-tryptase and contains a premature stop codon that results in the loss of the C-terminal regions essential for normal catalytic process. *TPSD1* has been implicated in the pathogenesis of autoimmune diseases [19,20]. Even though *TPSD1* is largely inactive, there is a possibility that increased expression of this gene might be a sign of increased MC activity [21].

Here, we investigated the relationship between *TPSB2* and *TPSD1* expression and BMI in the ST of HOA patients.

## 2. Results

### 2.1. Synovial Levels of TPSB2 and TPSD1 by BMI

The study enrolled 46 patients radiographically diagnosed with HOA. The distribution of patients according to age and the proportion of patients according to Tönnis grade (2/3) were similar between the two groups (*p* = 0.732 and *p* = 0.167, respectively; Table 1). BMI in the overweight (OW) group was significantly higher than that in the normal-weight (NW) group (*p* < 0.001, Table 1). There was a significant increase in *TPSB2* expression in the OW group compared to the NW group (Figure 1A; *p* = 0.038). In addition, the expression of *TPSD1* in the OW group was higher than that in the NW group (Figure 1B; *p* = 0.040).

### 2.2. Correlation between TSPB2 and TPSD1 Expression and Proinflammatory Cytokines

Proinflammatory cytokines such as TNF-α, IL-1β, and *IL6* play a pivotal role in synovial inflammation [22,23]. To estimate the possible association of *TPSB2* and *TPSD1* in synovial inflammation, correlation between their expression and inflammatory cytokine expression was investigated. *TPSB2* expression levels positively correlated with *TNFA* (r = 0.692, *p* < 0.001) and *IL6* (r = 0.302, *p =* 0.049) expression levels (Figure 2A,B). There was no difference between *TPSB2* and *IL1B* expression levels (r = 0.228, *p =* 0.142; Figure 2C). Similarly, *TPSD1* expression levels positively correlated with *TNFA* (r = 0.721, *p* < 0.001) and *IL6* (r = 0.333, *p =* 0.029) expression levels (Figure 2D,E). No correlation was observed between *TPSD1* and *IL1B* (r = 0.109, *p =* 0.486, Figure 2F).

### 2.3. Tryptase Expression Levels in MCs Derived from the NW and OW Groups

Clinical characteristics of patients determined by magnetic isolation are shown in Table 1. Comparison of the MC-rich fractions (MC-RFs) and MC-poor fractions (MC-PFs) using magnetic beads showed higher expression levels of *TPSB2* and *TPSD1* in the MC-RF than the MC-PF in both NW (*TPSB2*, *p* = 0.014; *TPSD1*, *p* = 0.021) and OW (*TPSB2*, *p* = 0.017; *TPSD1*, *p* = 0.138) (Figure 3A,B, respectively). There were no significant differences in the expression levels of *TPSB2* or *TPSD1* in the MC-RF and MC-PF between the NW and OW groups (*TPSB2*, *p* = 1.000; *TPSD1*, *p* = 1.000)

## 3. Discussion

In this study, we showed that *TPSB2* and *TPSD1* were significantly increased in HOA patients with OW. This increase in overweight patients may partly explain the link between BMI and HOA.

β-tryptase plays a pivotal role in the inflammatory process through the stimulation of inflammatory cytokine production. β-tryptase stimulates IL-1β production in synovial macrophages and fibroblasts derived from KOA patients [13]. β-tryptase increased TNF-α and *IL6* production in microglia and peripheral mononuclear cells [24,25]. A recent study also reported that β-tryptase modulated joint lubrication in OA through the cleavage of lubricin [10]. In contrast, recent studies have suggested that increased *TPSD1* gene expression levels were observed in several diseases [26,27]. For example, *TPSD1* expression was increased in patients with aspirin-exacerbated respiratory disease [26]. Increased *TPSD1* expression was also observed in patients with regenerative rotator cuff tear [27]. In our study, *TPSB2* and *TPSD1* positively correlated with *TNFA* and *IL6*. Increased *TPSB2* and *TPSD1* expression may be associated with inflammation in HOA in patients who are overweight.

The obese condition changes cell phenotypes and response to inflammatory stimulation [11,28,29]. Synovial fibroblasts from obese KOA patients had a high capacity to produce *IL6* protein [11]. Synovial fibroblasts in obese patients highly expressed transcriptional regulators MYC and FOS [29]. Synovial fibroblasts from obese KOA patients exhibited greater aerobic glycolysis, basal lactate secretion, and mitochondrial respiration when stimulated with TNF-α, compared to synovial fibroblasts from subjects with normal weight [28]. Bulk analysis of synovial tissue revealed that synovial *TPSB2* and *TPSD1* expression levels were increased in overweight HOA patients compared to normal-weight patients. We hypothesized that this result reflects the fact that MC-derived OW patients highly expressed *TPSB2* and *TPSD1*, and although *TPSB2* and *TPSD1* expression levels in MCs were similar between normal and overweight patients, an increased ratio/number of MCs results in an increase in *TPSB2* and *TPSD1* in ST of overweight patients. We isolated MCs from normal and overweight patients but found no difference between the groups in *TPSB2* and *TPSD1* expression. These results may suggest that elevated *TPSB2* and *TPSD1* expression in ST of overweight HOA patients may reflects an increased MC number or ratio in ST of overweight patients. In contrast, basophils also express *TPSB2* and *TPSD1* [30]. Therefore, elevated *TPSB2* and *TPSD1* expression in overweight HOA patients may reflect an increase in non-MC populations expressing these genes. Further investigation regarding MC number or MC ratio using flow cytometric analysis or immunohistochemistry is needed to clarify this hypothesis.

Several limitations of the present study warrant mention. First, sample size was small. Second, we did not examine a control population. Inclusion of normal and overweight HOA populations in the study is essential to confirm whether *TPSB2* and *TPSD1* expression is increased in HOA with overweight. Third, we analyzed mRNA expression in the synovium. Further investigation to complement these findings is necessary, such as a protein-profiling study using Western blot as a complementary measure to our gene expression results. Finally, the relationship between *TPSB2* and *TPSD1* expression and synovial inflammation or OA pathology remains to be determined.

## 4. Material and Methods

### 4.1. Patients

The study was conducted with a retrospective design in patients radiographically diagnosed with HOA at our institution between 2020 and 2022. Inclusion criteria were age ≥40 years and symptomatic primary HOA (Tönnis classification system II–III).

We excluded patients with previous hip surgery, current or previous use of immunosuppressive medication, and HOA caused by rapidly destructive coxarthritis, idiopathic osteonecrosis of the femoral head, pigmented villonodular synovitis, trauma, or rheumatoid arthritis. Samples were extracted from ST lining the anterior joint capsule in the inferior part of the femoral neck adjacent to the femoral head during the total hip arthroplasty procedure via the anterolateral supine approach in each subject. The subjects were grouped according to the World Health Organization’s BMI classifications, namely normal weight (NW; <25 kg/m^2^) and overweight (OW; 25–29.99 kg/m^2^). Expression of synovial tryptase (*TPSB2*, *TPSD1*) mRNA was investigated by real time PCR. Tissues from the remaining two sets of 14 patients (normal, *n* = 7 and overweight, *n* = 7) were used to compare tryptase gene expression between mast cells derived from normal and overweight patients.

### 4.2. Clinical Assessment

From the radiographic assessment of the progression of HOA, the following grades were assigned based on the Tönnis classification system [31]. Patient background factors and clinical evaluations were compared between patients with normal weight and overweight following quantitative PCR (qPCR) analysis and MC isolation, as described in Table 2.

### 4.3. qPCR

RNA extraction, cDNA synthesis, and qPCR were conducted using methods previously described in literature [14]. Briefly, primers used for qPCR are listed in Table 2. Gene expression was normalized to that of GAPDH using the delta–delta Ct method. Relative expression was calculated using the mean of all samples (synovial samples from the NW group or MC-poor fraction from the NW group).

### 4.4. Magnetic Isolation of MC

MC isolation was performed using magnetic isolation methods reported previously [13]. Briefly, cells were obtained from collagenase-digested synovial samples derived from NW and OW patients. After centrifugation, cells were reacted with biotin-labelled antibody cocktail (anti CD3, CD14, CD19, CD90). All antibodies were purchased from Biolegend (San Diego, CA, USA). Following reaction with streptavidin-conjugated magnetic beads (BD™ IMag Streptavidin Particles Plus—DM, BD Biosciences, Tokyo, Japan), the MC-RF were isolated by negative selection. Positive fractions were also isolated as an MC-PF. *TPSB2* and *TPSD1* in the MC-RF and MC-PF were estimated by qPCR instrument (Bio-Rad CFX Connect, Bio-Rad, Hercules, CA, USA).

### 4.5. Statistical Analysis

We used non-parametric analyses, as continuous variables in all analyses showed a non-normal distribution by the Shapiro–Wilk test. Continuous and categorical variables between the two groups were compared using the Mann–Whitney U-test and Pearson’s chi-squared test, respectively. Gene expression between the groups was compared using the Kruskal–Wallis test adjusted with Bonferroni correction. All statistical analyses were conducted using commercial software (IBM SPSS Statistics for Windows version 28, IBM Corp., Armonk, NY, USA), with p values of less than 0.05 considered to indicate statistical significance.

## 5. Conclusions

*TPSB2* and *TPSD1* expression is increased in synovium of overweight HOA patients. Further investigation is needed to reveal the role of tryptase in the relationship between increasing BMI and HOA pathology.

## Figures and Tables

**Figure 1 ijms-24-11532-f001:**
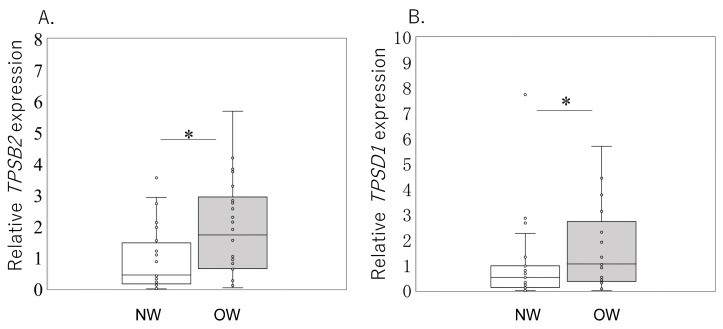
Expression of *TPSB2* and *TPSD1* in the synovial tissue of normal-weight (NW) and overweight (OW) patients. The expression of *TPSB2* and *TPSD1* mRNA in NW and OW groups was estimated by qPCR (**A**,**B**). *TPSB2* (**A**) and *TPSD1* (**B**) mRNA expression in the synovial tissue of normal-weight (*n* = 25) and overweight (*n* = 21) patients with hip osteoarthritis. Gene expression is presented in box and whisker plots, showing the median, 25th, and 75th percentiles and range. * *p* < 0.05.

**Figure 2 ijms-24-11532-f002:**
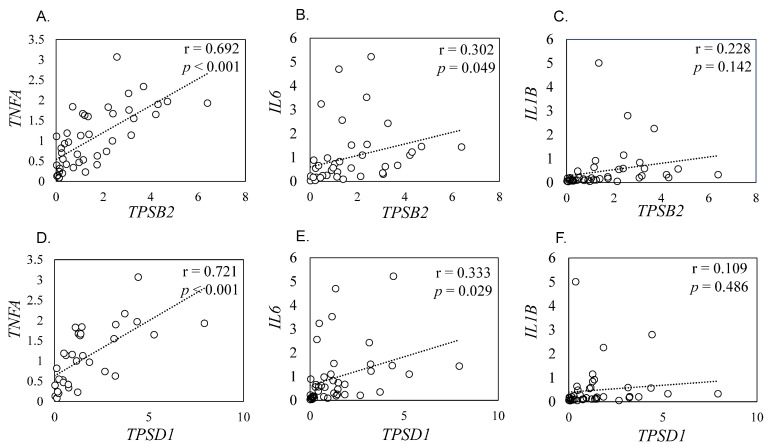
Correlation between *TPSB2* and *TPSD1* expression and proinflammatory cytokines expression. (**A**–**C**) Correlation between *TPSB2* and *TNFA* (**A**), *IL6* (**B**), and *IL1B* (**C**). (**D**–**F**) Correlation between *TPSD1* and *TNFA* (**D**), *IL6* (**E**), and *IL1B* (**F**).

**Figure 3 ijms-24-11532-f003:**
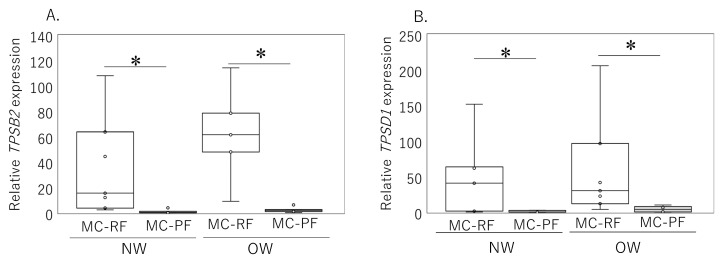
Expression of *TPSB2* and *TPSD1* in mast-cell-rich fraction (MC-RF) and mast-cell-poor fraction (MC-PF) obtained from normal-weight (NW) and overweight (OW) patients. The expression of *TPSB2* (**A**) and *TPSD1* (**B**) mRNA in MC-RF and MC-PF (NW, *n* = 7; OW, *n* = 7) was estimated by qPCR. Gene expressions are presented in box and whisker plots, showing the median, 25th, and 75th percentiles and range. * *p* < 0.05.

**Table 1 ijms-24-11532-t001:** Clinical characteristics of normal and overweight patients with hip osteoarthritis.

	qPCR			Magnetic Isolation		
	NW (*n* = 25)	OW (*n* = 21)	*p*	NW (*n* = 7)	OW (*n* = 7)	*p*
Age (years)	62.6 ± 9.8	63.0 ± 12.9	0.732	60.6 ± 13.1	58.4 ± 7.8	0.456
Tönnis (2/3), *n*	8/17	11/10	0.167	4/3	2/5	0.383
BMI (kg/m^2^)	21.9 ± 2.2	27.1 ± 1.4	<0.001	22.6 ± 1.34	26.5 ± 1.3	<0.001

NW, normal weight, OW, overweight, BMI, body mass index.

**Table 2 ijms-24-11532-t002:** Sequences of the primers used in this study.

Gene	Direction	Primer Sequence (5′–3′)	Product Size (bp)
*TPSB2*	F	CGCAAAATACCACCTTGGCG	138
	R	GTGCCATTCACCTTGCACAC	
*TPSD1*	F	CGGAATATCACACCGGCCTC	135
	R	TGCCATTCACCTTGCAGACC	
*TNFA*	F	CTTCTGCCTGCTGCACTTTG	118
	R	GTCACTCGGGGTTCGAGAAG	
*IL1B*	F	GTACCTGTCCTGCGTGTTGA	153
	R	GGGAACTGGGCAGACTCAAA	
*IL6*	F	GAGGAGACTTGCCTGGTGAA	199
	R	TGGCATTTGTGGTTGGGTCA	
*GAPDH*	F	TGTTGCCATCAATGACCCCTT	202
	R	CTCCACGACGTACTCAGCG	

## Data Availability

The data presented in this study are available on request from the corresponding author.

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
