# Peer review of "Increase in TPSB2 and TPSD1 Expression in Synovium of Hip Osteoarthritis Patients Who Are Overweight"

_ijms, 2023, doi:10.3390/ijms241411532_

Round 1

Reviewer 1 Report

Introduction 

·         Line 32:  it should be added that OA is a whole joint disease.

·         Line 36: the involvement of synovial membrane in OA should be introduced.

·         Genes should be written in italics.

·         line 50 wich triyptase?

Results 

·         At the beginning of the results the authors should report the number of patients enrolled rather than in the methods.

·         The authors compared the gene expression of two genes first  in patients NW vs OW and then isolated MC from the patients and compared the expression. They found an increase of expression of these two genes in OW vs NW but no difference was found when comparing the expression of the same genes in MC isolated from these patients. Why? In the introduction it is stated that MCs produce these cells thus this point in unclear and deserve to be discussed. Moreover, it would be important to quantify the number of isolated MCs in NW vs OW patients.

·         The authors should evaluate the inflammatory state of synovial membrane as it is well-known that this tissue is inflamed in OA and this could affect the results. 

Discussion

·         Lines 93-94: this part is unclear. The authors did not quantify the number of MC cells.

·         Lines 99-100: pathology should be specified.

·         Lines 112-114: it should be demonstrated.

Methods

·         Inclusion criteria should be specified.

·         Lines 133-136: the authors should better explain how they collected synovial membrane and in which part of the joint.

·         Lines 140-141: the authors should move table 2 in the qPCR section.

·         Lines 145-146: The authors cite reference 11 for the protocol. However, if you read reference 11, there is no protocol. It is written to read other 2 papers. It would be better to clarify the protocol.

·         Lines 150-151: the authors cite ref 11 and 25. However, if you read these papers there is written to see other papers. Thus, the protocol should be added.

·         Supplier of reagents should be added. Instruments used should be added.

·         SPSS should be correctly cited as reported by the website of the software.

Other comments

·         Abbreviations should be defined (line 45-47) and used consistently throughout the manuscript. ST should be defined.

·         Lines 180-184: this part should be filled.

·         Since the study involves patients, Ethical Committee approval number and data is mandatory.

Author Response

Response: Thank you for reviewing our manuscript and your valuable comments

Comments and Suggestions for Authors

Introduction

  • Line 32: it should be added that OA is a whole joint disease.

Response: We have added “whole” in Line 32.

  • Line 36: the involvement of synovial membrane in OA should be introduced.

Response: We have introduced the involvement of the synovial membrane in OA with a new citation (line 35-37)

  • Genes should be written in italics.

Response: Genes have been italicized in the text.

  • line 50 which triyptase?

Elevation of tryptase enzyme activity in synovial extract of KOA patient was reported in reference 17. However, the subtype remains unclear. We have moved the sentence to line 46-47.

Results

  • At the beginning of the results the authors should report the number of patients enrolled rather than in the methods.

Response: The sentence has been moved from the Methods to the Results section. (line 62)

  • The authors compared the gene expression of two genes first in patients NW vs OW and then isolated MC from the patients and compared the expression. They found an increase of expression of these two genes in OW vs NW but no difference was found when comparing the expression of the same genes in MC isolated from these patients. Why? In the introduction it is stated that MCs produce these cells thus this point in unclear and deserve to be discussed. Moreover, it would be important to quantify the number of isolated MCs in NW vs OW patients.

Response: Bulk analysis of synovial tissue revealed that synovial TPSB2 and TPSD1 expression levels were increased in overweight HOA patients compared to normal weight patients. We hypothesized that this result reflects 1) MC derived from OW patients highly expresses TPSB2 and TPSD1, or 2) TPSB2 and TPSD1 expression levels in MC were similar between normal and overweight patients, but the increased ratio/number of MC results in an increase in TPSB2 and TPSD1 in ST of overweight patients. We have added these points in the Discussion section.

As you suggest, quantifying the number of isolated MCs was important. We have added this point in the Discussion section. (line 111-116)

  • The authors should evaluate the inflammatory state of synovial membrane as it is well-known that this tissue is inflamed in OA and this could affect the results.

Response: We did not evaluate inflammatory state. We have added this point in the Discussion section as a limitation. (line 129-130)

Discussion

  • Lines 93-94: this part is unclear. The authors did not quantify the number of MC cells.

Response: We have revised this sentence to avoid overreaching.

  • Lines 99-100: pathology should be specified.

Response: We have revised this sentence. (line 100)

  • Lines 112-114: it should be demonstrated.

Response: As number of MC were not demonstratred, we have revised this sentence.

Methods

・ Inclusion criteria should be specified.

We have added inclusion criteria in the Methods section (line 134 –135)

  • Lines 133-136: the authors should better explain how they collected synovial membrane and in which part of the joint.

We have added the details in the Methods section (line 139-141)

  • Lines 140-141: the authors should move table 2 in the qPCR section.

Response: We have moved Table 2 in the qPCR section.

  • Lines 145-146: The authors cite reference 11 for the protocol. However, if you read reference 11, there is no protocol. It is written to read other 2 papers. It would be better to clarify the protocol.

Response: We apologize for this mistake.  We have cited reference 14.

  1. Tsukada, A.; Takata, K.; Takano, S.; Ohashi, Y.; Mukai, M.; Aikawa, J.; Iwase, D.; Inoue, G.; Takaso, M.; Uchida, K., Increased NMUR1 Expression in Mast Cells in the Synovial Membrane of Obese Osteoarthritis Patients. Int J Mol Sci 2022, 23, (19).

  • Lines 150-151: the authors cite ref 11 and 25. However, if you read these papers there is written to see other papers. Thus, the protocol should be added.

Response: We have established MC isolation methods using magnetic beads in Reference 11 (new reference 13; details are written in the section “Expression of TPSB2 and IL1B in MC” in the Patients and Methods section). In reference 11, there are no references to other papers.

As suggested, because reference 25 included text requiring the reader to refer to other papers, we have removed reference 25.

  • Supplier of reagents should be added. Instruments used should be added.

Response: We have added the information in the Methods section.

  • SPSS should be correctly cited as reported by the website of the software.

Response: We have correctly revised it based on the website of the software.

Other comments

  • Abbreviations should be defined (line 45-47) and used consistently throughout the manuscript. ST should be defined.

Response: ST has been defined in line 36.

  • Lines 180-184: this part should be filled.

Response: We have included the Institutional Review Board Statement, Informed Consent Statement, and Data Availability Statement. (line 195-202)

  • Since the study involves patients, Ethical Committee approval number and data is mandatory.

Response: We have added this information. (line 196)

Reviewer 2 Report

Dear Authors,

I read your manuscript with interest and appreciation. Every research dealing with hip osteoarthritis etiology is very important because the disease leads to serious disability and usually with the need of operative treatment. Unfortunately, as you mentioned there is a serious methodological limitation of the research: the lack of a control group. In my opinion you should compare your results of both overweight and non-overweight hip osteoarthritis patients with two control groups: overweight and non-overweight participants without hip osteoarthritis. This may influence both the results and conclusions, which may be different in comparison to the present study. 

Author Response

I read your manuscript with interest and appreciation. Every research dealing with hip osteoarthritis etiology is very important because the disease leads to serious disability and usually with the need of operative treatment. Unfortunately, as you mentioned there is a serious methodological limitation of the research: the lack of a control group. In my opinion you should compare your results of both overweight and non-overweight hip osteoarthritis patients with two control groups: overweight and non-overweight participants without hip osteoarthritis. This may influence both the results and conclusions, which may be different in comparison to the present study.

Response: Thank you for reviewing our manuscript. We agree with the reviewer’s opinion that analysis of non-HOA patients is very important. We have added this point in the Limitations section. (line 124-126) However, obtaining synovium from non-HOA patients is difficult in Japan due to ethical problems. Our original manuscript had several limitations and is insufficient as an original article. However, we have carefully revised the manuscript based on reviewers’ comments and believe that it is now suitable for publication as a communication article.

Round 2

Reviewer 1 Report

It is still unclear why the authors found a difference comparing gene expression in synovial tissues and then, this difference disappears when isolating the cells. It is important to clarify this point. It is likely that other cells contribute to the expression of the genes.

The authors should evaluate synovial inflammation level.

These two points are important for the quality of the manuscript.

Author Response

Thank you for reviewing our revised manuscript and providing important suggestions. As reviewer’s suggested, TPSB2 and TPSD1 expression were observed in non-MC populations, basophils. Therefore, other cells contribute to the elevation of TPSB2 and TPSD1 in overweight patients. We have added this point in discussion section. (Line 138-140)

We also evaluated inflammatory cytokine expression including TNFA, IL1B, and IL6 to estimate synovial inflammation levels. We found that TPSB2 and TPSD1 expression correlated with TNFA and IL6 (new Figure 2). We have added this information in results and discussion section. (Line 79-92; Line 113-114; Line 120)

Reviewer 2 Report

Dear Authors,

I would like to thank you for the response and changes in the manuscript. In the present form your paper is ready for publication as a communication article.

Author Response

Thank you for reviewing our revised manuscript.

Round 3

Reviewer 1 Report

No additional comments